# Consensus structure prediction of A. thaliana's MCTP4 structure using prediction tools and coarse grained simulations of transmembrane domain dynamics

**Sujith Sritharan**[1], **Raphaelle Versini**[1], **Jules D Petit**[2], **Emmanuelle E Bayer**[2], **Antoine Taly**[1]*

**1** Laboratoire de Biochimie Théorique, CNRS (UPR9080), Université Paris Cité, Paris, France,
**2** Laboratoire de Biogenèse Membranaire, Unité Mixte de Recherche 5200, Université de Bordeaux, Centre National de la Recherche Scientifique, Villenave d'Ornon, France

* taly@ibpc.fr

**Data availability statement:** All of the analysis and data used in this study have been documented and made available for reference

## Abstract

Multiple C2 Domains and Transmembrane region Proteins (MCTPs) in plants have been identified as important functional and structural components of plasmodesmata cytoplasmic bridges, which are vital for cell-cell communication. MCTPs are endoplasmic reticulum (ER)-associated proteins which contain three to four C2 domains and two transmembrane regions. In this study, we created structural models of *Arabidopsis* MCTP4 ER-anchor transmembrane region (TMR) domain using several prediction methods based on deep learning (DL). This region, critical for driving ER association, presents a complex domain organization and remains largely unknown. Our study demonstrates that using a single deep-learning method to predict the structure of membrane proteins can be challenging. Our models presented three different conformations for the MCTP4 structure, provided by different deep learning methods, indicating the potential complexity of the protein's conformational landscape. We then used physics-based molecular dynamics simulations to explore the behaviour of the TMR of MCTPs within the lipid bilayer. We found that the TMR of MCTP4 is not rigid but can adopt multiple conformations. The membrane-embedded region contains two helical pairs: HP1 (TM1–TM2) and HP2 (TM3–TM4). Deep learning predictions revealed three distinct types of inter-helical contact interfaces: ESMFold, AlphaFold-Multimer, trRosetta, and RoseTTAFold consistently predicted a TM2–TM3 contact; AlphaFold2 did not predict any contact between these two helical pairs, while OmegaFold instead suggested a TM1–TM4 interface. Our physics-based coarse-grained simulations not only confirmed the contacts predicted by these models but also revealed a broader conformational landscape. In particular, structural clustering identified five distinct conformational clusters, with additional and more extensive inter-helical contacts not captured by the deep learning predictions. These findings underscore the complexity of predicting

and reuse. These scripts are hosted in Jupyter Notebooks, a popular open-source web application that allows for the creation and sharing of documents. The notebooks can be accessed from the associated GitHub repository at https://github.com/Jouffluu/ Molecular-modelling-MCTP.

**Funding:** This work was supported by the European Research Council (ERC) under the European Union's Horizon 2020 research and innovation program (project n° 772103-BRIDGING to EB), labex DYNAMO (11-LABX-0011 to AT) and ANR projects DIVCON (ANR-21-CE13-0016-01 to AT) and MITOFUSION (ANR-19-CE11-0018 to AT).

**Competing interests:** No authors have competing interests.

protein structures. We learned that combining different methods, such as deep learning and simulations, enhances our understanding of complex proteins.

## 1 Introduction

Plasmodesmata (PD) are intercellular channels found in plants that allow for the communication and transport of molecules between adjacent cells [1]. PD have a unique membrane organization characterized by tight membrane contact sites, consisting of two concentric membranes - the plasma membrane (PM) and the endoplasmic reticulum (ER) [1]. The regulation of intercellular trafficking through these channels is essential for plant growth, development, and defense against biotic and abiotic stresses [2].

The ER-PM tethering machinery of membrane contact sites in plasmodesmata has been hypothesised to play a crucial role in PD formation, reshaping, and proper function [1,2]. Members of the MCTP family are plasmodesmata-localised and act as ER-PM tethers [3]. MCTPs localize at plasmodesmata and function as tethering proteins, with their transmembrane region anchored in the endoplasmic reticulum and their C2 domains interacting with the plasma membrane. They are thought to play a crucial role in regulating intercellular signaling [2].

MCTP proteins consist of two transmembrane regions and three or four tandem C2 domains. The C2 domains act as PM docking sites through interaction with anionic sites, while the transmembrane domain insert into the ER membrane. This structural organization is essential for their function in PD. The cytosolic C2 domains were recently shown, through both Martini simulations and experimental evidence, to interact with PI4P lipids. These interactions support a role for MCTP4 in regulating intracellular trafficking via its C2 domains [4]. In parallel, the transmembrane region (TMR) has been identified as critical for stabilizing ER connections at plasmodesmata, with mutations in this region disrupting ER-PM continuity and leading to pore closure [5]. Despite this functional importance, the TMR remains structurally unresolved, and its three-dimensional organization is poorly characterized.

MCTP proteins are expressed in various species. Invertebrate organisms such as *Caenorhabditis elegans* and *Drosophila melanogaster* express a single MCTP gene, whereas vertebrates express two MCTP genes (MCTP1 and MCTP2) [6]. In *Arabidopsis thaliana*, there are 16 members of this family. However, no complete experimental structure of any MCTP is currently available in the literature [7].

Recently, a variety of new protein structure prediction tools have emerged, namely Alphafold and RosettaFold, as well as new strategies based on large language models, including ESMFold and OmegaFold [8–11]. These tools are powerful and trained to recognize evolutionary preserved structural motifs. However, they are also very recent, and we are just beginning to understand their limitations. Unlike AlphaFold, which relies heavily on multiple sequence alignments (MSA) and experimentally determined structures from the Protein Data Bank (PDB), ESMFold and Omegafold leverages a large-scale language model trained on vast and diverse sequence datasets, including metagenomic sequences for ESMFold. This approach makes both models less dependent on the availability of experimentally solved structures, such as those of membrane proteins. These methods employ the protein sequence, potentially generating more accurate predictions for membrane proteins.

In the absence of an experimentally determined structure for MCTP4, we employed a prospective approach by comparing predictions from multiple deep learning-based tools (Alphafold(AF), Alphafold multimer(AFM), RosettaFold(RF), Tr-RosettaFold(TR), ESMFold(ESM), and OmegaFold(OF) [8–13] ) to explore its conformational landscape.

Convergence across these models, combined with MD simulations, was used to mitigate this risk. Finally, we employed a two-tiered molecular dynamics (MD) simulation strategy: (1) Martini 3 coarse-grained simulations for extensive conformational sampling of the transmembrane domain, and (2) nine independent Charmm36 all-atom simulations (three starting structures with three replicas each) to verify structural stability and evaluate potential helix extrusion events. This complementary approach was coupled with principal component analysis to characterize the conformational landscape. We examine the structure and temporal flexibility of the TMR, as well as its behavior within a lipid bilayer. We find that MD explores the conformational space sampled by DL tools and beyond.

## 2 Material and methods

### 2.1 Predicted models

The three-dimensional structure of the full-length *Arabidopsis thaliana* MCTP4 also called FT-interacting protein 4 (Uniprot: Q9C8TM3) was obtained using deep-learning prediction tools. We used Alphafold (version 2.2) [8], Alphafold multimer (version 2.2) [12], Omegafold (version 1.1.0) [10], and ESMfold (version 1.0.3) [14], which were run on a local cluster. For Rosettafold, we used the public webserver (https://robetta.bakerlab.org/submit.php) [9], and for Tr-rosetta, we used the webserver (https://yanglab.nankai.edu.cn/trRosetta) [13]. We used the AF3 web server (https://alphafoldserver.com/) to generate additional predictions, while Boltz-1 and Chai-1 models were run on a local server [15,16].

### 2.2 Computational detail

All simulations, both Coarse-Grained (CG) and All-Atom (AA), were performed with GROMACS 2021.5 simulation package, with system setup facilitated by the CHARMM-GUI webserver [17]. For each simulation type, we initiated the process with an energy minimization phase, followed by a series of equilibration steps, before proceeding to the production simulations.

### 2.3 System preparation

We utilised the Martini maker from the CHARMM-GUI web server for the construction of the system and the mapping of atomistic structures to the CG Martini 3 models [18]. Our simulations targeted residues 550 to 776, encompassing 50 residues preceding the transmembrane domain, the transmembrane (TM) helix domains (600 to 750), and extending to the end of the protein. The precise lipid composition of the ER at plasmodesmata remains unknown, but the ER membrane in plants is well-characterized as primarily PLPC/PLPE [19]. To investigate the stability of the TMR helix, we therefore performed simulations in PLPC/PLPE (80:20) bilayer membranes for AA-MD and PIPC/PIPE environments for CG-MD [5], as a reasonable approximation of the ER lipid environment anchoring MCTP4. Each model was oriented using the PPM server by incorporating the topology of the N-terminus of the first chain from the PDB file. Subsequently, each system was solvated in water, neutralized, and supplemented with 0.15 M NaCl. Additionally, we created all-atom systems using the membrane builder from CHARMM-GUI [20], following the same process with the CHARMM36 force field, and constructed the membranes with equivalent lipids.

### 2.4 CG-MD simulations

In all simulations, GROMACS 2021.5 simulation package was used with the Martini 3 force field [21,22]. The protocol consisted of an initial energy minimization phase, followed by a

multi-stage equilibration process, and concluded with a production simulation. Energy minimization was carried out for two iterations of 5,000 steps each using the steepest descent method. The simulations were subsequently equilibrated through five stages, employing time steps of 2, 5, 10, 15, and 20 fs. A target temperature of 300 K was maintained with the v-rescale thermostat, with a coupling constant of 1 ps. An semiisotropic pressure of 1 bar was maintained using the Parrinello-Rahman barostat [23], with a compressibility of $4.5 \times 10\text{-}5$ bar-1 and a relaxation time constant of 12 ps. Long-range interactions were treated with a cutoff radius of 1.1 nm for both van der Waals and Coulombic interactions, using a switching function from 1.0 nm for van der Waals. The production simulations were performed using an NPT ensemble with a time step of 20 fs for a total simulation time of 3 microseconds. Multiple replicates were conducted for the system, and the final phase of the simulation was executed with no restraints.

## 2.5 AA-MD simulations

In our AA-MD (All-Atom Molecular Dynamics) simulations, we constructed three distinct transmembrane systems (AF, ESM, OF) using the CHARMM36m force field via CHARMM-GUI. To mimic the environment of plant endoplasmic reticulum, we opted for a lipid composition of 20% PLPE and 80% PLPC. A Verlet cutoff scheme was employed for interaction handling. Van der Waals interactions were addressed with a cutoff and force-switch at 1.0 nm, while electrostatic interactions were managed via the Particle Mesh Ewald method, both set to a cutoff of 1.2 nm. Pressure coupling was achieved using the Parrinello-Rahman method with a semiisotropic type.

## 2.6 Contact analysis

To quantify contacts between transmembrane (TM) helices among the different models predicted by the algorithms, we used the MDAnalysis library to calculate distances between the centers of mass of the residues [24,25]. During the simulations, a distance map from 0 to 12 angstrom was used to analyze distance between the membrane-embedded region containing two helical pairs: HP1 (TM1–TM2) and HP2 (TM3–TM4). This threshold was selected to account for the resolution of coarse-grained Martini models, where inter-residue contact typically span longer distances than in all-atom representations, and to ensure that relevant helix–helix interactions were not missed.

## 2.7 Principal component analysis and clustering

To reveal the most important motions in the TM helices, we employed principal components analysis (PCA) using the tools provided in the GROMACS software package [21]. For all models, we first fitted the trajectories where the transmembrane (TM) helix part was stable within the membrane for 3 microseconds to ensure that the phosphates of the membrane remained in the same position. We then concatenated these adjusted trajectories. The concatenated trajectories were further supplemented with three additional models derived from AF3, Boltz-1, and Chai-1. The covariance matrix was calculated on the backbone atoms of the residues located in the membrane (HP1 and HP2). The first two principal components were then plotted using the matplotlib library for visualization [26]. The results of the PCA were clustered using the K-Medoids method. Utilizing the KMedoids class from the scikit-learn-extra library [27], we specified five clusters for categorization. We determined this optimal number of clusters using the elbow method. After fitting the model to the PCA data, the coordinates of the cluster centroids were determined.

## 2.8 Data analysis and visualisation

All of the analysis and data used in this study have been documented and made available for reference and reuse. These scripts are hosted in Jupyter Notebooks, a popular open-source web application that allows for the creation and sharing of documents. The notebooks can be accessed from the associated GitHub repository at https://github.com/Jouffluu/Molecular-modelling-MCTP.

Some basic analysis tasks, such as the calculation of the root mean square deviation (RMSD) or the distances between the centres of the geometry of the two helices, were performed using the tools provided in the GROMACS 2021.5 software package [21]. For visualisation purposes, we constructed a density graph to provide a view of the distribution of data points within the first two main components (PC1 and PC2). VMD [28] was used to visualise trajectories and ChimeraX was used to analyse alphafold output [29].

## 3 Results

### 3.1 Models of MCTP4 ER-TMR domain

We generated six models of MCTP4 ER TMR domain using six different prediction methods (S4 Fig), in addition to a partial model of MCTP4 TMR domain previously constructed by Modeller and used as a reference [7]. This later work utilized bioinformatic tools, hydrophobic clusters analysis and molecular dynamics, to delineate transmembrane helices within the MCTP4 protein [7]. We decided to use the resulting definition of membrane domains. Therefore, we extracted the 550-776 region from each model. We previously identified five putative subdomains within the approximately 200-residue sequence of MCTP4 TMR. These subdomains include an N-term amphipathic helix (APH1), a putative transmembrane domain (TMD0), a hairpin transmembrane domain (HP1) composed of two transmembrane helices (TM1 and TM2), a second, longer amphipathic helix (APH2), and another hairpin transmembrane domain (HP2) which is also composed of two transmembrane helices (TM3 and TM4) [7]. These subdomains are illustrated in Fig 1.

Subsequently, an alignment and Root Mean Square Deviation (RMSD) calculation were performed on the extracted regions and on the whole model. The results of this analysis are shown in Table 1 and Fig 3g to 3i.

The PLDDT score was used to assess the confidence of the 3D structure prediction algorithms. A low PLDDT score indicates low confidence in the 3D structure, while a high score indicates higher confidence. Confidence scores varied among the models, particularly in the membrane region, as illustrated in S3 Fig. Among the prediction methods, ESM appeared to be the most confident in its predictions for both TM domains, while AlphaFold (AF2) seemed to be less confident in the HP2 domain compared to the other methods, see Fig 2. The other methods showed relatively high confidence in their predictions.

When comparing the contact maps, which are two-dimensional representations of three-dimensional structures [31,32], of the models predicted by each method, we observed that ESM, TR, RF, and AFM predictions showed a similar pattern, see Fig 3a to 3b and Fig 3d to 3e. In these conformations, the TM2 and TM3 helices (615-635, 715:735) were found to be in close proximity. On the other hand, the models made by AF2 and OF displayed different conformations in the membrane domain. In AF2's model, the TM2-TM3 helices were separated by more than 8 angstroms with almost no contact see Fig 3e and 3i. Furthermore, it's root mean square deviation (RMSD) was high, over 13 angstrom, compared to other models see Table 1. This shows that AF2's model was quite different in terms of structure.

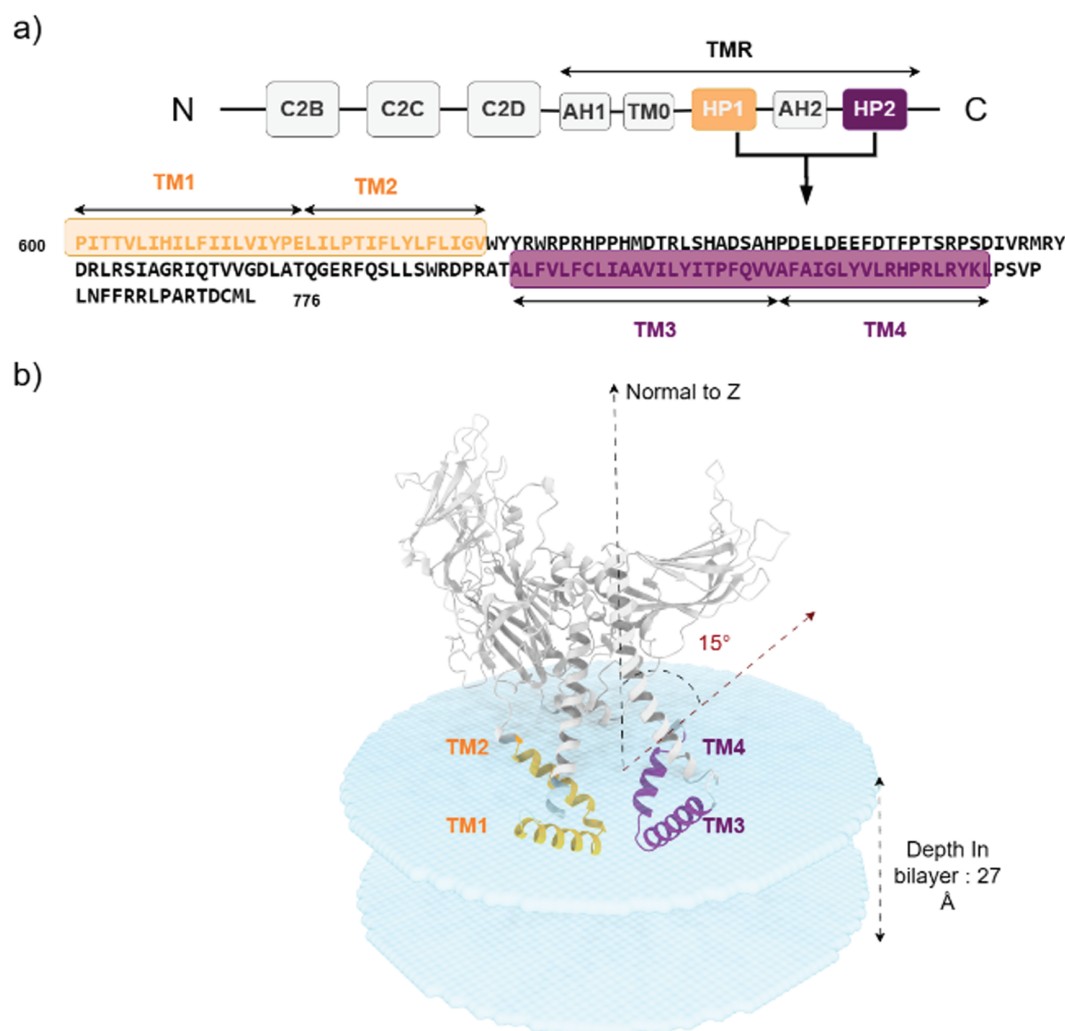

**Fig 1. a) MCTP4 TMR having 5 elements: 2 transmembrane hairpins (HP1 and HP2) and 3 amphipathic helices (AH1, TM0 and AH2).** b) AF MCTP4 model oriented in the lipid bilayer by OPM server [30].

**Table 1. RMSD values for the transmembrane domain (550-776) (upper half) and for the alpha carbon of entire models (lower half). For each pair of models, the RMSD value is indicated for each measurement.**

|  | AF | AFM | Rose | Tr-rose | Esmfold | Omegafold |
|---|---|---|---|---|---|---|
| AF | 0 | 13.23 | 14.29 | 13.55 | 14.28 | 15.63 |
| AFM | 8.03 | 0 | 6.18 | 3.07 | 5.57 | 15.67 |
| Rose | 18.04 | 16.58 | 0 | 6.56 | 8.13 | 16.17 |
| Tr-rose | 8.75 | 4.43 | 17.16 | 0 | 4.95 | 15.37 |
| Esmfold | 16.11 | 15.13 | 19.28 | 13.35 | 0 | 16.13 |
| Omegafold | 26.40 | 27.00 | 27.00 | 27.37 | 31.33 | 0 |

When looking at the conformation obtained by OF, the close connection was between helices TM1 and TM4 (residues 600:615, 735:750), as seen in Fig 3f and 3h. OF's model also had a high RMSD, around 15, compared to the other models (Table 1). These large RMSD values and different helix arrangements highlight the range of protein conformations predicted by different methods.

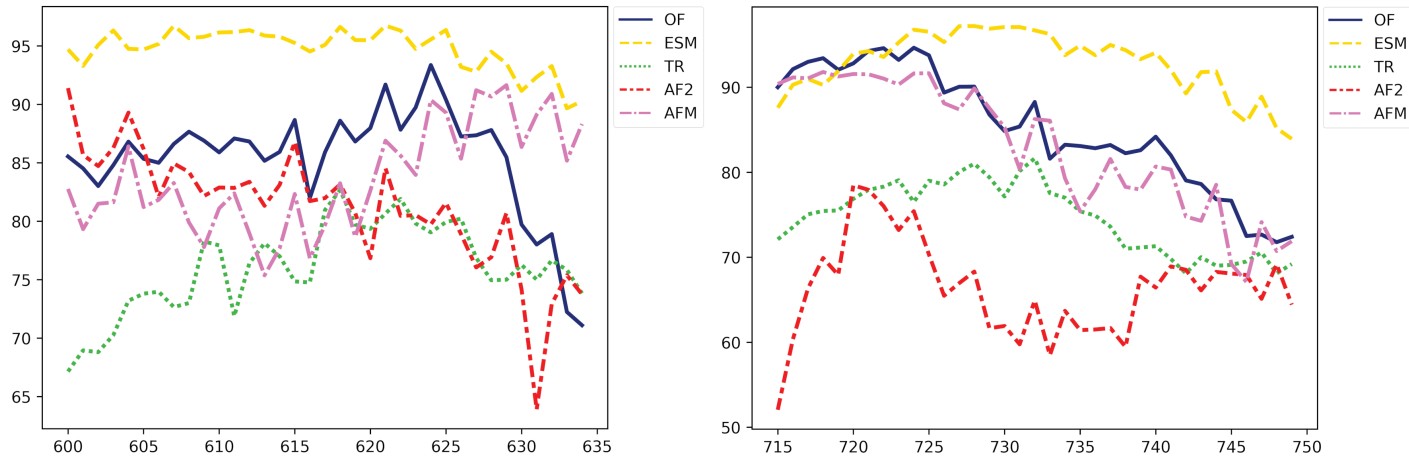

**Fig 2. Evaluation of model predictions using the pLDDT score as a function of residues in the HP1 (left) and HP2 (right) regions.** The curves of different colors represent models predicted by various prediction methods: AlphaFold (AF, red), OmegaFold (OF, blue), TR (green), ESM (yellow), and AlphaFold Multimer (AFM, magenta).

**Dimers.** Because other MCTP have been proposed to dimerize [33], we wanted to determine whether the ER-domain of MCTP4 has the potential to form dimers. The analysis of the dimer produced by Alphafold multimer revealed that intra-subunits contacts were similar to those found in the ESM, RF, and TR models, between TM2 and TM3. In terms of inter-subunits contacts, the TM1 helix of one monomer and the TM4 helix of the other one were found to interact Fig 4a and 4c). Furthermore, the algorithm exhibits high confidence in the intra-domain but also in the inter-subunits-interactions Fig 4b and 4d.

In every model, the MCTP4 TMR incorporates two hairpins predicted to be in the bilayer membrane. While four of the models converge on similar arrangements and distances between the hairpins, two models provide notably different configurations. To investigate the stability of each model and whether they would interconvert, we studied them using molecular dynamics simulations.

## 3.2 Molecular dynamics simulations

We performed coarse-grained molecular dynamics simulations with the Martini 3 force field, to investigate the behaviour of the TMR domains of each model, produced by the deep learning methods, included in a lipid bilayer that mimics the composition of the ER, as detailed in the materials and methods section. Ten replicates were conducted for each model, each lasting 3 microseconds. During the initial simulations, we observed that certain TM domains (HP1 or HP2 or both) exited the membrane, which could potentially introduce variability in our analyses. For examples of HP1 and HP2 domain emergence from the membrane and related variability, see S7 and S9–S14 Figs. To ensure consistency, we decided to exclude these simulations from our analysis. We then repeated the simulations until we obtained four simulations for each starting model whose TM domains remained inside the membrane throughout the entire simulation. For RMSD data of these CG simulations, see S15 Fig. This approach allowed us to obtain a consistent dataset for further analysis.

Interestingly, out of the four simulations conducted for each of the AF2 and ESM models, one simulation from each model exhibited significant variations in the distances between the TM2 (HP1) and TM3 (HP2) helices during the simulation, with both proximity and

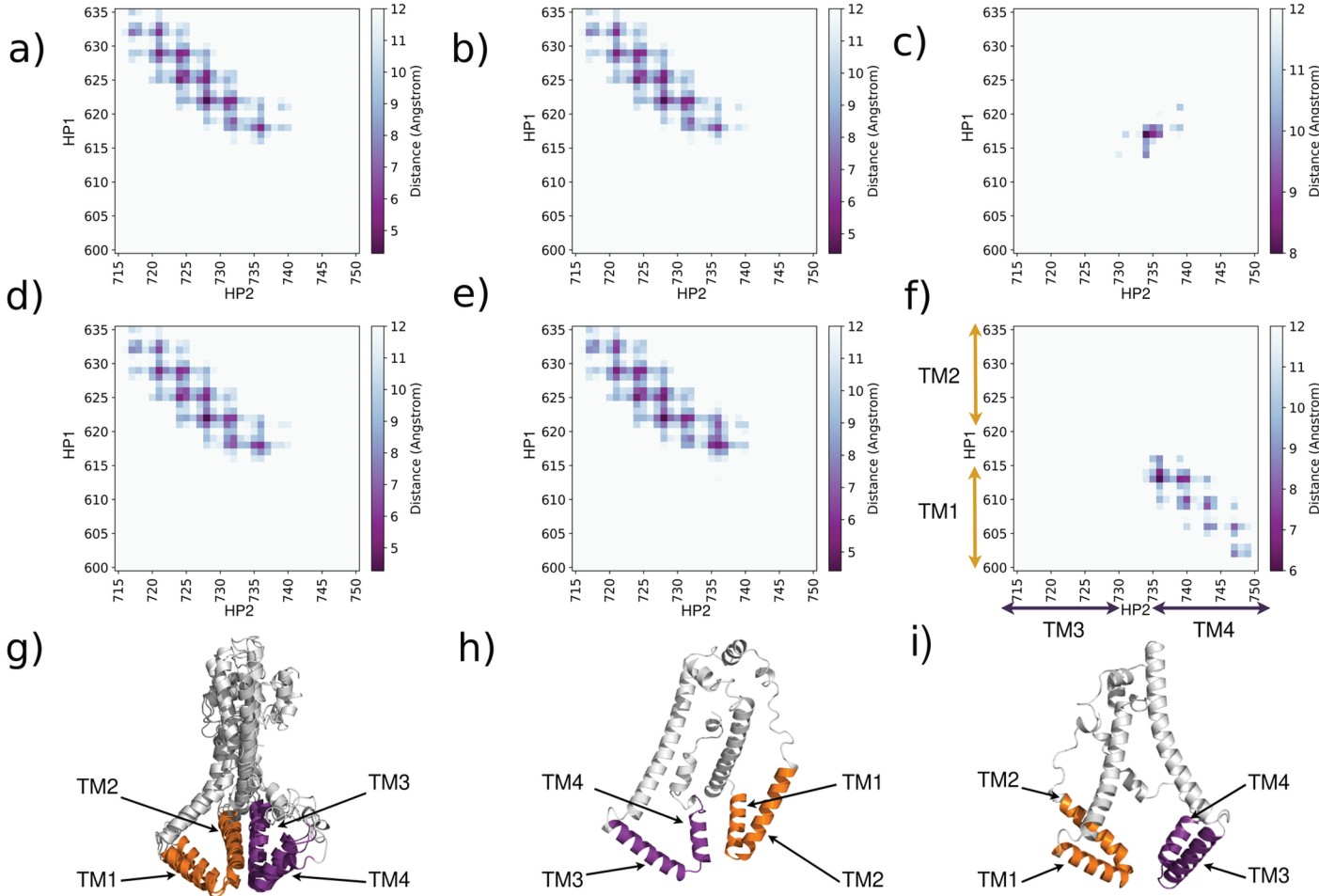

**Fig 3. Contact maps between the residues of transmembrane HP1 and HP2 of different protein structures: a) ESM, b) AFM, c) AF, d) TR, e) RF and f) OF.** The distances were calculated using the alpha carbon atoms. Heat map coloring indicates the distance between residues, ranging from dark purple for short distances to white for longer distances(12 A). g) superposition of ESM, AFM, TR and RF transmembrane domain models. h) and j) are OF and AF transmembrane domains.

separation observed. Despite these movements, these particular simulations demonstrated stability within the membrane, indicating that the helices were not confined to a single conformation (See S8 Fig for distance measurements between HP1 and HP2 domains in AlphaFold CG simulations). This also indicates that the system did not exhibit ballistic motion, but rather stochastic dynamics consistent with thermal fluctuations, suggesting that the simulations may have reached equilibrium.

To gain a deeper understanding of the dynamic behaviour of these helices within the membrane, we conducted further characterization using principal component analysis, to get insights into the conformational dynamics of TM domains in lipid bilayer membranes.

### 3.3 Principal component analysis and clustering

We employed principal component analysis (PCA) as a statistical method to elucidate the most significant motions of the TM helices within the lipid bilayer. In PCA, the Cartesian coordinates (X, Y, Z) of each atom were used as descriptors to capture the accessible degrees of freedom of the protein. Specifically, we selected the residues of the TM helices located in

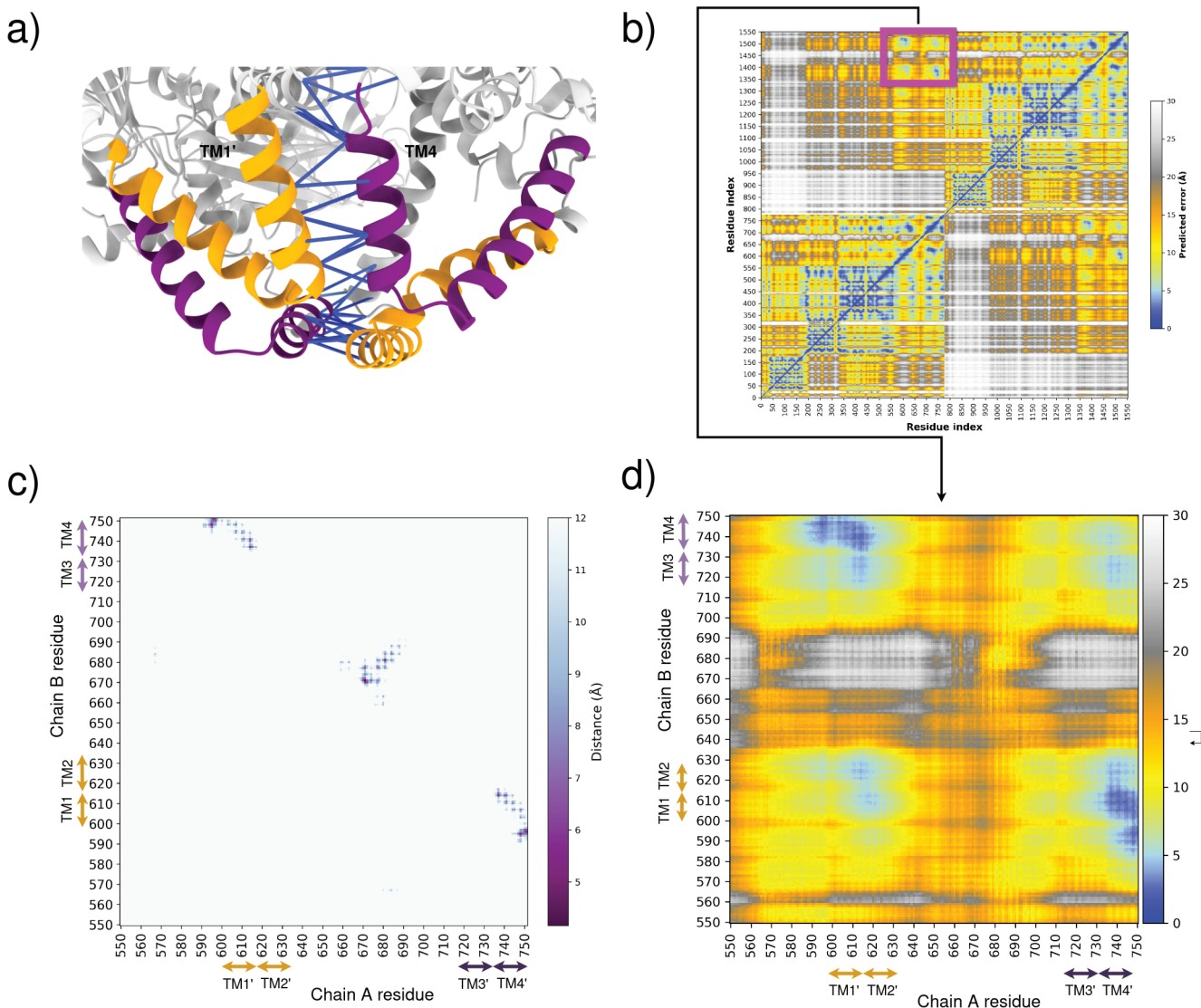

**Fig 4. a) AlphaFold multimer prediction of TMR MCTP4 protein.** The HP1 domain is represented in orange and the HP2 domain in purple. The pLDDT score between residues of interchain domains is represented by lines, with the blue color indicating a high score. b) Predicted Aligned Error (PAE) for the MCTP4 dimer, extracted by AlphaPickle [34]. PAE provides an estimate of the error in the predicted alignment of residues. c) Inter-chain contact map of TMR. d) Close-up view of the PAE for the TMR of MCTP4.

the membrane to perform the PCA. The resulting conformations from the simulations were projected onto the first two principal components, which accounted for 49% of the variability observed in our simulations (see Fig 5a). For detailed frequency and projection analyses, see S5 and S6 Figs.

Each data point in the figure represents a conformation from the trajectory of an MCTP TMR model obtained through different methods, with the respective starting points of each model also displayed. Despite distinct starting points for each model, the conformations generated by the simulations appear to convergence towards two distinct basins (see Fig 5c). The main basin regroups the 6 similar models (ESM, RF, TR, AFM, Boltz and Chai) and most of the structures including some coming from simulations started from AF2 and OF. The second

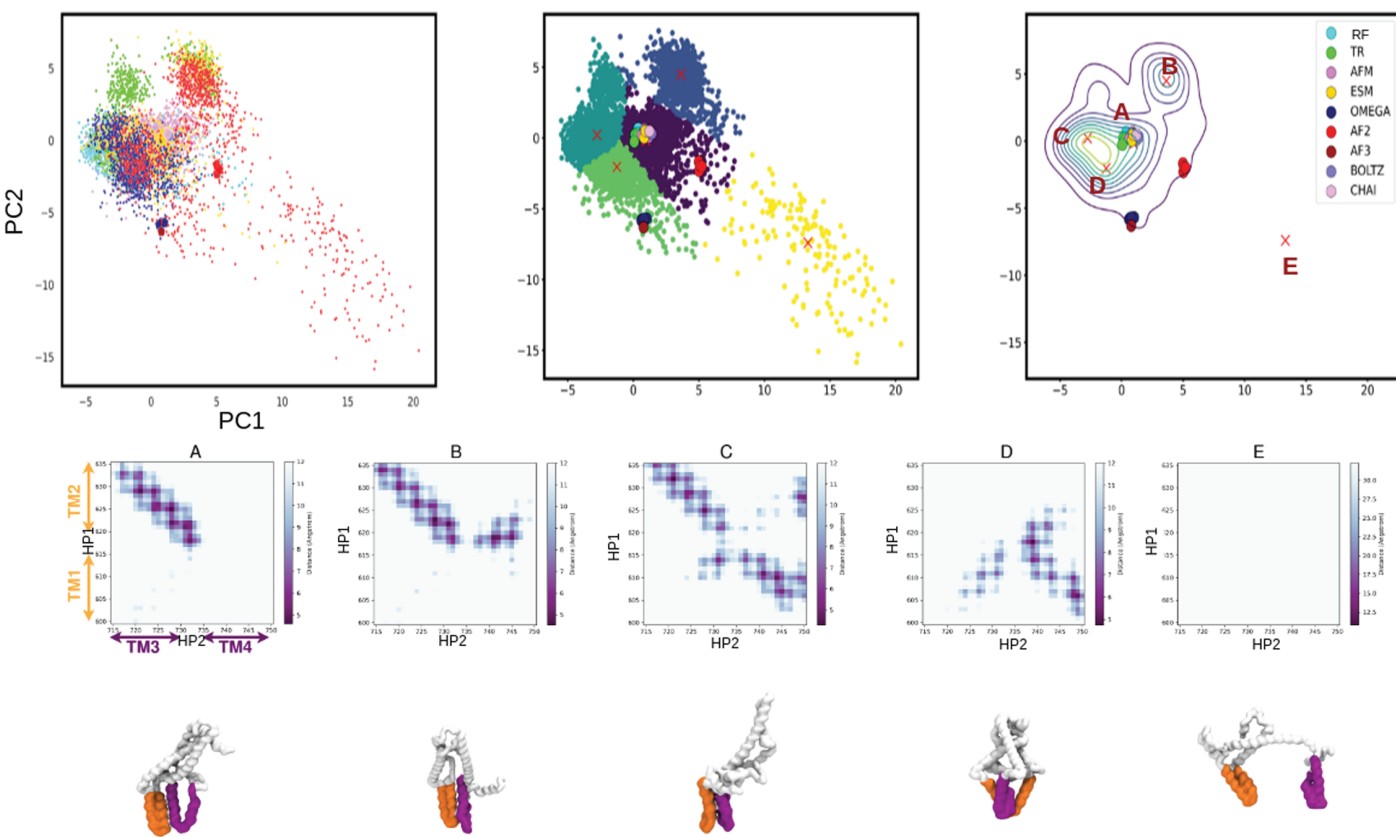

**Fig 5. Principal Component Analysis (PCA) plots, clustering and density representation.** a) The plot shows a projection based on a PCA's first two principal components (PCs). Large points represent the starting points of each simulation for each model, and small points are conformations generated by the simulation in each model. Each model is represented by its own colour: AlphaFold (AF, red), OmegaFold (OF, blue), TR-rosetta (TR, green), ESMFold (ESM, yellow), AlphaFold Multimer (AFM, magenta), AlphaFold v3 (AF3, dark red), Boltz-1 (Boltz, light purple) and Chai-1 (Chai, light pink). b) The centroids of each basin have been determined using the K-medoid clustering method. c) A density plot showing the projection of the two PCs, where each point corresponds to the starting point of each model. d) The representative structure of each cluster is shown with a contact map corresponding to contacts between transmembrane domains.

basin does not include any of the starting structures but is formed by structures extracted from the simulations started from various models (AF2, ESM, RF and TR). To complement and validate the PCA results, we also performed Time-lagged Independent Component Analysis (TICA), which captures the slowest modes of motion over time. TICA confirmed the separation into two major groups, consistent with the PCA projection. Additionally, TICA revealed that the AF models explore three distinct directions in conformational space, suggesting heterogeneous dynamic behavior not fully resolved by PCA alone (see S18 and S19 Figs).

To complement our analysis, we included three additional models (AF3, Boltz-1, and Chai-1) in the PCA without performing MD simulations on these structures. Principal Component Analysis (PCA) revealed that each all-atom replica was firmly anchored in its own conformational space, without significant overlap between different spaces.

To identify the centers of these basins, we performed clustering using the k-meloids algorithm on the two principal components extracted from PCA. This allowed us to identify five distinct clusters within the conformational space see Fig 5b. From each cluster, we extracted representative structures that best represented the characteristics of that particular cluster.

For the first cluster (Cluster A), we obtained a representative structure that closely resembled the initial "consensus" starting point of the simulation. This cluster is part of the main basin, suggesting that the simulation explored a conformational space similar to the initial structure throughout most of the simulation time.

In the second cluster (Cluster B), we obtained a representative structure that exhibited significant structural changes compared to the initial starting points. In this structure, helices TM2 are in contact with both TM3 and TM4 helices, which is novel and not captured by deep learning methods (see Fig 5d). This suggests that the simulation explored a different conformational space and underwent substantial structural rearrangements.

In the third cluster (Cluster C), which is also part of the main basin, the representative structure shows contacts corresponding to the TM1-TM4 and TM2-TM3 helices. This indicates a unique conformation not present in the initial models, further highlighting the simulation's ability to explore alternative conformations.

In the fourth cluster, (Cluster D), the representative structure illustrates contacts between TM1-TM3 and TM1-TM4 helices. This suggests a distinct conformation, emphasizing the diversity of conformations accessible to the system.

Finally, in the fifth cluster (Cluster E), the representative structure corresponds to a conformation where the TM domains are separated (Fig 5d). This suggests that this simulation explored a wider range of conformational space, further highlighting the dynamic nature of the system under study. The PCA and clustering analysis have provided deeper insights into the structure and dynamics of the MCTP4 TMR. Our findings indicate that the MCTP4 TMR is not rigid, but displays substantial potential for structural rearrangement within the bilayer. There's a prominent convergence towards two main conformational basins, signifying two key structural states that the TMR can adopt. Furthermore, we identified certain novel helical contacts, implying that the MCTP4 TMR can explore a broad conformational space. This underscores the dynamic and versatile nature of the MCTP4 TMR in its biological context.

**3.3.1 Atomistic simulations for enhanced analysis.** To complement and reinforce our analyses based on coarse-grained simulations, we also carried out atomistic simulations of the TMR domains. These simulations were performed on the three forms of TMR domains (ESM, AF and OF) obtained through deep learning methods, aiming to test the stability and reliability of the predicted conformations. For each conformation, three replicas were run, each lasting 250 nanoseconds and Root Mean Square Deviation (RMSD) analyses were performed (see S2 Fig). These analyses showed that the RMSD remained below 0.5 nm for the simulations started from models obtained with OmegaFold and ESMFold. Taken together with the coarse-grained simulation this observation confirms their relative stability. These analyses showed that the RMSD remained below 0.5 nm for the simulations initiated from models obtained with OmegaFold and ESMFold, indicating their relative structural stability. This observation is consistent with the coarse-grained simulations, which also suggested stable behavior for these models. In contrast, the simulations started from the AlphaFold2-derived model exhibited higher RMSD values, reaching approximately 0.8 nm in the all-atom simulations. This increased deviation supports the coarse-grained results and suggests that this conformation is less stable, potentially due to a reduced number of contacts between HP1 and HP2 (see Fig 3).

## 4 Discussion

Our study revealed that the MCTP4 TMR domain, located within the ER, consists of two hairpin, each containing two helices. These hairpin are deeply embedded within the lipid

bilayer of the ER. The arrangement of these hairpins and their proximity to each other varied, reflecting the complexity and dynamism of the MCTP4 TMR structure within an ER-mimicking bilayer.

## 4.1 Exploration of the conformational landscape by deep learning models

In this study, we generated six distinct models of MCTP4 TMR using various prediction methods, which allowed us to explore the conformational landscape of the protein. Notably, PLDDT scores revealed varied confidence levels in the predictions of MCTP4's TMR 3D structure, with ESM showing the highest confidence in its predictions for both TM domains. It is interesting to note that four of the prediction methods (ESM, TR, RF, and AFM) converged on a similar MCTP's transmembrane domain, where helices TM2 and TM3 are in close proximity. This suggests that this particular conformation might be a reliable representation of the protein's actual structure. However, AF2 and OF produced distinct models with different helical arrangements. Additional predictions from AF3 showed similarities with the OF model, while Boltz-1 and Chai-1 aligned more closely with the consensus group (ESM, TR, RF, and AFM), further supporting the existence of these two main conformational patterns. These divergences could indicate the flexibility of MCTP4 TMR and the presence of alternative conformations. These findings highlight the value of using multiple structure prediction methods in parallel to explore a protein's conformational landscape. Combining predictions leverages the complementary strengths of different tools, reduces the impact of individual errors, and provides a more reliable view of the likely structural states [35].

## 4.2 Molecular dynamics simulations

**4.2.1 Simulations behavior with martini 3.** During our simulations, we investigated the dynamic behaviors of the TM helices of MCTP4 within a lipid bilayer membrane designed to mimic the composition of the endoplasmic reticulum. With Martini 3, we were able to observe TM Helices that show spatial flexibility. Nevertheless, among the initial 10 replicas for each model, we observed a diverse range of helix behaviors within the membrane. Some helices were coming out of the membrane (S9–S14 Figs). Consequently, we repeated the simulations multiple times to obtain a consistent and stable set of results within the membrane. In order to ensure the reliability of our analysis, we excluded simulations in which the TM domains exited the membrane. The Martini 3 force field has been developed in part to solve the issue identified with martini2, that tends to overestimate protein-protein interactions [22] However, Martini3 is conversely associated with too hydrophilic alpha-helices. Recent studies have shown that Martini 3 tends to favor adsorbed states over transmembrane (TM) states for short peptides, due to overly strong protein–water interactions. This leads to membrane ejection events that are not observed in higher-resolution simulations. Umbrella sampling confirms that Martini 3 lowers the energy barrier for TM-to-adsorbed transitions and stabilizes the adsorbed state by 20 kJ/mol. Notably, once a helix exits the membrane, it does not reinsert, consistent with our observations [36]. A similar issue was also reported in recent work, where the shorter version of a TM helix (PEPT1-29) exited the membrane, preventing dimerization, while a longer variant (PEPT1-41) showed improved TM stability [37]. In our case, we observed such exits occurring at different times (early, mid, or late in the trajectory) and across different models, including those initially inserted correctly. To assess whether this instability was a true feature of the protein or a force field artifact, we ran nine independent all-atom simulations (three different models × three replicas each), none of which showed TM helix ejection. This strongly supports the interpretation that the ejections observed in Martini 3 are not meaningful, but rather artifacts of the force field. We therefore excluded

those Martini simulations from our analysis, not to suppress variability, but to avoid drawing conclusions from nonphysical behavior. Importantly, we retained and analyzed several stable Martini simulations (i.e., those in which all TM helices remained embedded), which consistently sampled conformational diversity within the membrane environment, without restraints or bias. In summary, our exclusion of membrane ejection cases is justified by physical consistency with higher-resolution models and recent literature identifying limitations in Martini 3's treatment of TM peptide stability.

**4.2.2 Combining all-atom and coarse-grained MD.**   Principal Component Analysis (PCA) revealed that each all-atom replica was firmly anchored in its own conformational space, without significant overlap between different spaces. This observation underlines that the replicas did not explore other conformational regions, indicating a clear segregation and distinction between the conformations of each TMR domain form (see S1 Fig).

These atomistic results complement our coarse-grained simulations by demonstrating that the TMR domain forms, while being stable and distinct within their respective conformational spaces (see S1 Fig), do not visit other potential conformations. This highlights the advantage of coarse-grained simulations in exploring a wide array of conformations, thus offering a broader perspective on the conformational dynamics of proteins in membrane environments, complemented by the precision of atomistic simulations to validate and deepen our understanding of specific molecular dynamics.

Combined, these approaches provide a comprehensive and detailed view of the dynamics and stability of TMR domains within the membrane, thereby enriching our understanding of the underlying mechanisms governing the behaviour of MCTP4 protein in a biomembrane context.

**4.2.3 The plus of molecular dynamics run on top of deep-learning models.**   Three starting points were generated by deep learning methods and for each model, 4 times 3 $\mu$s simulations were launched. This allowed the exploration of a conformational landscape with two basins. The center of the first basin is very close to the conformation majorly found by deep learning algorithms. However, the other clusters present representative structures with contacts that are not found in deep learning methods, suggesting that with MD simulation, we were able to explore new conformations which are not found by deep learning methods. This is coherent with the observation that even the most recent diffusion-based methods do not manage to explore the full conformational space in all cases [38]. It's also interesting to see that the AF2 and OF models start right at the edge of the consensus basin and then move into it. This highlights the value of combining physics-based MD simulations with deep learning models, allowing us to leverage their complementary strengths for more robust structural insights.

## 4.3 Monomer versus dimer contacts

Analyzing the predictions of deep learning models, we have observed significant differences between the TMR models of AF2, OF, and those that have converged, namely RF, AFM, TR, and ESM.

In the TMR model of AF2, the two helices of each transmembrane hairpin do not come into contact. This feature stands in stark contrast with those observed in other models. In the TMR model from OF, it's noteworthy that helices TM1 and TM4 are in contact. Similarly, the converged models exhibit a distinct contact between helices TM2 and TM3. This analysis thus reveals a variety of structural configurations predicted by different deep learning models, underscoring the complexity of transmembrane interactions of MCTP4.

Through the use of coarse-grained molecular dynamics simulations, we found that the interaction between TM2 and TM3 tended to be more stable. This observation is supported by our density map, which reveals two distinct basins. Models that display these interactions are predominantly found in the larger of the two basins.

In contrast, the models produced by AlphaFold (AF) and OpenFold (OF) are positioned away from these basins. This positioning suggests that these models may represent an intermediate state. Further insights can be gleaned from studying the dimeric form of the protein.

Interestingly AF2 and AFM provided different results. This prompted us to explore further the difference. Noteworthy, AF2 showed significant deviations and obtained a low PLDDT score, particularly in the TM regions. This implies that AlphaFold's monomeric predictions are not always reliable or accurate for certain protein domains. The variation observed between monomeric and multimeric predictions could indicate that the formation of the dimeric structure involves additional interactions or structural rearrangements not captured in the monomeric prediction. A key capability of AF2 is to allow the prediction of contacts from sequence alignments [39,40]. The relationship between sequences and contacts is however partially ambiguous, which has been shown in the case of conformational changes. This in turn triggered the creation of strategies to explore the conformational landscape with AF2 and RoseTTAFold [41–46]. It is therefore tempting to speculate that evolutionary signals are not necessarily captured by AlphaFold's monomeric models for membrane proteins that form homo-oligomers. In the case of AF2, the model might be subject to conflicting constraints corresponding to intra-subunit and inter-subunit contacts. We further predicted oligomeric assemblies of MCTP4 to investigate whether the structural features observed in monomeric models were maintained or reorganized upon oligomerization. Notably, the TM2–TM3 contacts identified in the consensus monomeric models (ESM, TR, RF, AFM) were preserved as intra-subunit interactions in the predicted dimers. In addition, AlphaFold-Multimer (dimer and trimer) revealed a simultaneous inter-subunit interface involving TM1 from one subunit and TM4 from the adjacent subunit. In contrast, the alternative TM1–TM4 interface observed in the OmegaFold monomer model was recovered in the oligomeric prediction as an inter-chain contact only, with no corresponding intra-subunit TM2–TM3 interaction. This supports the idea that the consensus monomeric conformation captures both intra- and inter-subunit constraints relevant to oligomeric assembly, whereas OmegaFold may reflect only partial features of the full oligomeric interface see Fig 4.

The model produced by OF shows contacts between TM1 and TM4 helices that rather appear to be inter-subunit contacts with AFM. This is coherent with the notion that there are conflicting constraints, based on coevolution, corresponding to intra-subunit and inter-subunit contacts in homo-oligomers, in particular using a monomeric prediction tool such as OF. Interestingly, the exploration of this model with a physics-based method like molecular dynamics could help resolve these conflicting constraints.

Predicting the structure of multimers is particularly challenging, as it involves assessing inter-subunit interfaces, which adds complexity compared to monomeric predictions. This has sparked discussion on whether protein interfaces should be modeled independently, as proposed by Zhu and colleagues [47], who suggest that dedicated treatment of interfaces could improve prediction accuracy. More broadly, this raises the question of whether multimeric proteins should be modeled directly as oligomers rather than as isolated monomers, since evolutionary constraints and structural signals may be embedded within their interface regions. Furthermore, this leads to the question of whether multimeric proteins should be modeled as monomers or oligomers. Oligomeric proteins could benefit from more precise treatment as oligomers rather than monomers, as part of the evolutionary pressure or signal might be associated with interfaces.

Our study demonstrates that the TMR of MCTP4 is structurally flexible and capable of adopting multiple conformations. By comparing several state-of-the-art deep learning structure prediction tools, we identified distinct inter-helical contact patterns, with a consensus around TM2–TM3 interactions in several models, and alternative TM1–TM4 contacts in others. Through physics-based coarse-grained simulations, we confirmed the stability of these predicted interfaces and uncovered additional conformational states not captured by static models alone. Structural clustering revealed five distinct conformational clusters, underscoring the dynamic nature of the TMR. These findings highlight the value of combining deep learning and molecular dynamics approaches to better understand the conformational landscape of membrane proteins, particularly for those involved in oligomerization and membrane tethering functions, such as MCTP4.

## Supporting information

**S1 Table. Statistical information on each model from OPM server.**
(PDF)

**S1 Fig. Projection of the first principal components (PC1 against PC2) from all atom simulation.** Each point represents an observation. Colors represent different models: Red for AF2, Navy Blue for OMEGA, and Gold for ESM.
(TIF)

**S2 Fig. RSMD from all-atom simulation.** Red for AF2, Navy Blue for OMEGA, and Gold for ESM.
(TIF)

**S3 Fig. Evaluation of model predictions using the pLDDT score.** The curves of different colors represent models predicted by various prediction methods: AlphaFold (AF, red), OmegaFold (OF, blue), TR (green), ESM (orange), and AlphaFold Multimer (AFM, purple).
(TIF)

**S4 Fig. Panels A to F (AF, AFM, OF, ESM, TR): Structures are colored according to pLDDT scores; blue signifies areas with high confidence (reliable structural prediction), while red depicts areas with low confidence (uncertain structural prediction).** Panel G displays the prediction made by RosettaFold, colored on the error estimate in the Å (RMSD) metric. This metric categorizes structural predictions into three levels of confidence: blue for very confident, red for less confident, and white for areas lacking confidence.
(TIF)

**S5 Fig. Models from AF3 (dark red), Boltz-1 (light purple) and Chai-1 (light pink).** The AF3 model shows structural TMR similarities with the Omegafold model, while Boltz-1 and Chai-1 align with the consensus conformations observed in ESM, AFM, TR, and RF models.
(TIF)

**S6 Fig. Variance explained by each Principal Component (PC) on CG simulations.**
(TIF)

**S7 Fig. Projection of the first principal components (PC1 against PC2, PC3 and PC4).** Each point represents an observation. Colors represent different models: Aqua for Rose, Lime Green for TR, Violet for AFM, Gold for ESM, Navy Blue for OMEGA, and Red for AF2.
(TIF)

**S8 Fig. Left: HP2 domain emerging from the membrane.** Center: HP1 domain emerging from the membrane. Right: Both HP1 and HP2 domains emerging from the membrane. These emergences from the membrane are observed in some replicas, regardless of the system in CG simulations. For our analysis, we chose not to consider these instances. Other predictive tools, such as PSIPRED [48] or DREAMM [49], indicate that this domain remains inside the membrane. When one or both domains emerge, they never re-enter the membrane during the simulation.
(TIF)

**S9 Fig. Distance between HP1 and HP2 domains throughout an AlphaFold CG simulation.**
(TIF)

**S10 Fig. Distance of the center of mass along the Z-axis for the HP1 and HP2 domains of the AF model.**
(TIF)

**S11 Fig. Distance of the center of mass along the Z-axis for the HP1 and HP2 domains of the AFM model.**
(TIF)

**S12 Fig. Distance of the center of mass along the Z-axis for the HP1 and HP2 domains of the RF model.**
(TIF)

**S13 Fig. Distance of the center of mass along the Z-axis for the HP1 and HP2 domains of the TR model.**
(TIF)

**S14 Fig. Distance of the center of mass along the Z-axis for the HP1 and HP2 domains of the OF model.**
(TIF)

**S15 Fig. Distance of the center of mass along the Z-axis for the HP1 and HP2 domains of the ESM model.**
(TIF)

**S16 Fig. RMSD for CG simulations.**
(TIF)

**S17 Fig. Dynamical cross-correlation analysis of TM helices in the concatenated trajectory.** The analysis reveals that the two helices within each transmembrane hairpin (HP1 and HP2) move as coherent units, showing positive correlation within each hairpin. Between HP1 and HP2, a weaker positive correlation is observed, consistent with the formation of cluster D identified in the PCA analysis.
(TIF)

**S18 Fig. TICA and PCA analysis.**
(TIF)

**S19 Fig. TICA with different lagtime and dimension.**
(TIF)

## Author contributions

**Conceptualization:** Emmanuelle E. Bayer, Antoine Taly.

**Data curation:** Jules D. Petit.

**Formal analysis:** Sujith Sritharan.

**Funding acquisition:** Emmanuelle E. Bayer, Antoine Taly.

**Investigation:** Sujith Sritharan, Raphaelle Versini, Emmanuelle E. Bayer, Antoine Taly.

**Methodology:** Antoine Taly.

**Supervision:** Antoine Taly.

**Writing – original draft:** Sujith Sritharan, Antoine Taly.

**Writing – review & editing:** Sujith Sritharan, Raphaelle Versini, Jules D. Petit, Emmanuelle E. Bayer, Antoine Taly.

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
