## [Decision Letter · Decision Letter 0]

7 Mar 2025

PONE-D-25-07701Prediction of A. thaliana′s MCTP4 Structure using Deep Learning-Based tools and Exploration of Transmembrane domain Dynamics using Coarse-Grained Molecular Dynamics SimulationsPLOS ONE

Dear Dr. Taly,

Thank you for submitting your manuscript to PLOS ONE. After careful consideration, we feel that it has merit but does not fully meet PLOS ONE’s publication criteria as it currently stands. Therefore, we invite you to submit a revised version of the manuscript that addresses the points raised during the review process.

The reviewers raised significant scientific and technical questions regarding the manuscript  "Prediction of A. thaliana′s MCTP4 Structure using Deep Learning-Based tools and Exploration of Transmembrane domain Dynamics using Coarse-Grained Molecular Dynamics Simulations". I am happy to consider the revised version after the major comments by the reviewers are sufficiently addressed. 

We look forward to receiving your revised manuscript.

Kind regards,

Soumendranath Bhakat

Academic Editor

PLOS ONE

Journal Requirements:

“This work was supported by the European Research Council (ERC) under the European

Union’s Horizon 2020 research and innovation program (project n° 772103-BRIDGING), labex DYNAMO (11-LABX-0011) and ANR projects DIVCON (ANR-21-CE13-0016-01) and MITOFUSION (ANR-19-CE11-0018).”

Reviewers' comments:

Reviewer's Responses to Questions

**Comments to the Author**

1. Is the manuscript technically sound, and do the data support the conclusions?

Reviewer #1: Partly

Reviewer #2: Partly

Reviewer #3: Yes

2. Has the statistical analysis been performed appropriately and rigorously? 

Reviewer #1: No

Reviewer #2: Yes

Reviewer #3: Yes

3. Have the authors made all data underlying the findings in their manuscript fully available?

Reviewer #1: Yes

Reviewer #2: Yes

Reviewer #3: Yes

4. Is the manuscript presented in an intelligible fashion and written in standard English?

Reviewer #1: Yes

Reviewer #2: No

Reviewer #3: Yes

5. Review Comments to the Author

Reviewer #1: The manuscript titled "Prediction of A. thaliana's MCTP4 Structure using Deep Learning-Based Tools and Exploration of Transmembrane Domain Dynamics using Coarse-Grained Molecular Dynamics Simulations" investigates the structural dynamics of the Arabidopsis thaliana MCTP4 protein. The study employs structure prediction methods and coarse-grained molecular dynamics simulations to analyze the transmembrane region (TMR) of MCTP4. This study advances the understanding of plant membrane protein structures and demonstrates the necessity of integrating multiple computational approaches for accurate structural modeling.

This paper requires major revisions before it can be considered for publication. The issues raised in the questions highlight key concerns regarding the interpretation of structural convergence, the handling of molecular dynamics results, and the reliability of deep learning-based predictions in the context of membrane proteins. The authors should address these points thoroughly and provide stronger justifications for their methodological choices. I recommend resubmitting the manuscript after a careful revision that clarifies these aspects.

Introduction

1.While deep learning plays a role in the structure prediction methods used, explicitly stating 'Deep Learning-Based Tools' in the title may be unnecessary. Since AlphaFold, RosettaFold, and similar tools are well-established in the field, I suggest simplifying the title to focus on the structural prediction and molecular dynamics aspects without emphasizing deep learning explicitly.

2.The authors mention that deep learning models such as Alphafold, RosettaFold, ESMFold, and OmegaFold have limitations due to the underrepresentation of membrane proteins in training datasets. However, the study relies on these tools to model the transmembrane region (TMR) of MCTP4. Given that deep learning models are inherently biased toward their training data, how do the authors quantitatively ensure that the models are not influenced by this bias? Have the authors benchmarked the results against experimentally solved membrane proteins of similar topology to assess reliability?

3.The study combines coarse-grained (Martini 3) and all-atom (CHARMM36) molecular dynamics simulations to validate the conformational dynamics of the predicted transmembrane region. However, coarse-grained models inherently reduce atomic detail, while all-atom simulations are computationally expensive and limited in timescales. How do the authors ensure that coarse-grained simplifications do not overlook key molecular interactions essential for membrane stability? Conversely, are the all-atom simulations long enough to capture biologically relevant conformational dynamics?

4.The introduction states that MCTP proteins act as ER-PM tethers and are localized at plasmodesmata, potentially regulating intercellular transport. However, the focus of this study is on predicting the structure of the MCTP4 transmembrane region. Given that the tethering function likely depends on interactions between the transmembrane and cytoplasmic domains, how does studying the isolated TMR provide meaningful insights into its function? Would not modeling the entire MCTP4 protein, including C2 domains, provide a more biologically relevant structural interpretation?

Materials and Methods

1.The authors state that they compared results from various deep learning-based prediction tools (Alphafold, RosettaFold, ESMFold, OmegaFold, etc.) to ensure reliability. However, the choice of models seems to rely primarily on convergence across tools, rather than on an objective experimental validation metric. Given that membrane proteins are underrepresented in training datasets for these models, how do the authors ensure that the observed convergence is not merely a reflection of shared biases across deep learning models rather than genuine structural accuracy? Would not benchmarking against experimentally solved membrane protein structures be a stronger validation approach?

2.The authors describe the construction of transmembrane systems in a PIPC/PIPE bilayer membrane for CG-MD and a PLPE/PLPC lipid environment for AA-MD simulations. However, the ER-PM contact sites in plasmodesmata may contain additional lipid diversity, such as sterols or specific phosphoinositides. How do the authors justify these lipid choices, and have they considered whether these specific membrane environments sufficiently mimic the in vivo conditions of MCTP4 anchoring? Would varying the lipid composition affect the stability and interactions of the TMR?

3.The study employs coarse-grained Martini 3 simulations for long-timescale behavior and all-atom CHARMM36m simulations for finer structural insights. However, Martini 3 force fields have been known to sometimes overestimate protein stability in membranes due to their implicit treatment of hydrogen bonding and electrostatics. On the other hand, the 3-microsecond all-atom simulations may not fully capture slow conformational changes. Given these known issues, how do the authors ensure that Martini 3 does not artificially stabilize the TMR, and how do they justify that their AA-MD simulations are sufficiently long to explore meaningful structural transitions?

4.The authors used PCA to analyze transmembrane motions and subsequently K-Medoids clustering to classify different states. However, PCA reduces the dataset to a linear representation, which may not fully capture the nonlinear nature of membrane protein dynamics. How do the authors justify using PCA for highly heterogeneous conformational sampling? Given that alternative methods such as time-independent component analysis (tICA) or Variational Autoencoders (VAEs) have been more effective in capturing slow conformational dynamics, why were such approaches not considered?

5.The authors state that they used VMD for trajectory visualization and ChimeraX for AlphaFold model analysis. However, visualization alone does not provide quantitative structural refinement. Given that homology-based models and MD-derived structures can contain local errors, why did not the authors perform explicit structure refinement using loop modeling, Rosetta relax, or MDFF (Molecular Dynamics Flexible Fitting)? Would not be necessary to ensure that the predicted TMR structures are physically plausible beyond deep learning model outputs?

Results

1.The authors state that their models exhibit significant structural variability, particularly in the arrangement of transmembrane helices. However, they also claim that PCA analysis converges towards two main basins. Given that some methods (e.g., OmegaFold) produce high RMSD values and distinct conformations, while others (e.g., ESMFold) cluster together, how do the authors ensure that this apparent convergence is not an artifact of the simulation constraints rather than a biologically meaningful state? Would longer AA-MD simulations potentially reveal alternative states not captured in the PCA clustering?

2.The study compares contact maps across models and finds that some models (e.g., ESMFold, RF, TR) share similar TM2-TM3 interactions, while others (AF2, OF) diverge significantly. However, contact maps only provide static pairwise interactions and do not inherently capture dynamic rearrangements. Given the high flexibility of MCTP4 TMR indicated by MD simulations, would not a time-resolved interaction network (such as correlation-based or dynamic cross-correlation analysis) provide a more robust measure of functionally relevant interactions? How do the authors justify relying on static contact maps when discussing model reliability?

3.The authors removed simulations in which TM domains exited the membrane to ensure consistency. However, excluding these simulations inherently introduces bias - potentially eliminating biologically meaningful states. Would not these deviations indicate that some models are intrinsically unstable within the lipid bilayer, suggesting that the underlying predicted structures might not be valid? Instead of excluding outlier simulations, why not analyze their membrane insertion free energy to determine whether these fluctuations are artifact-driven or biologically relevant?

4.The AA-MD simulations show that OmegaFold and ESMFold structures remain relatively stable, while AlphaFold2-derived models display higher RMSD values. However, the 250-ns timescale used for these simulations is relatively short, especially for membrane proteins where longer equilibration is often required. Given that coarse-grained models suggest high flexibility and multiple conformational states, how do the authors justify that 250 ns is sufficient to determine stability? Would not longer unbiased simulations or enhanced sampling methods (e.g., milestoning, weighted ensemble, or Markov state modeling) provide a more conclusive assessment of stability and conformational transitions?

5.It appears that several figures are referenced as "??" instead of the correct numbering. This suggests either missing figure captions or an issue with manuscript formatting. This should be carefully revised before submission, as it affects the readability and credibility of the results.

Discussion

1.The authors state that four models (ESM, TR, RF, AFM) converged on a similar MCTP4 transmembrane domain, specifically with TM2 and TM3 in close proximity. However, they also note that AF2 and OF produced distinct models and that AF3 showed similarities with OF, while Boltz-1 and Chai-1 aligned with the consensus group. How do the authors justify that the models in the so-called "consensus group" represent the biologically relevant conformation rather than a bias imposed by the prediction algorithms? Could the observed convergence be an artifact of training data or algorithmic assumptions rather than an actual structural feature of the protein?

2.The authors mention that some TM helices exited the membrane in coarse-grained simulations and that these cases were excluded from further analysis. Given that Martini 3 is known to exaggerate hydrophobic interactions, could the observed membrane exits be a sign of real instability rather than simulation noise? By removing these cases, is there a risk of overlooking biologically relevant conformations and underestimating the full conformational variability of the MCTP4 transmembrane domain?

3.The authors argue that AlphaFold's monomeric predictions may not fully capture evolutionary constraints associated with dimerization, as indicated by low PLDDT scores in TM regions. However, they also state that AFM identifies a strong inter-subunit contact between TM1 and TM4, and that OF predicts a TM1-TM4 contact but considers it an intra-subunit interaction instead. If AFM successfully predicts a dimeric interface, how do the authors conclude that evolutionary constraints are poorly captured? If some models recover these interactions, does this suggest that AlphaFold's handling of multimeric structures is more reliable than implied?

Reviewer #2: The language particularly in the presentation of the abstract, results and the discussion sections is rather not clear. They are overly informal or imprecise statements. To improve readability, the language should reworded in a more technical and scientific manner to strengthen the arguments presented in the manuscript and enhance readability.

Abstract

1. On the abstract, consider avoiding general statements and focus on specific results to convey the significance of the study to the reader.

Introduction

1. The sentence “These tethering proteins selectively concentrate at plasmodesmata, bind membranes together, and were hypothesised to control the exchange of information between cells” consider rewording the sentence for clarity.

Methods

1. A cutoff distance of 12 angstrom is used in the contact analysis between the beads “During the simulations, contacts between the beads representing the helices were defined using a cutoff distance of 12 angstrom, and contact occupancies were calculated for each bead pair.” Please clarify the reason for this threshold.

Results

1. Could you rewrite this sentence for clarity, “This also show that the system’s movement was not ballistic, which suggest that the simulations might have converged to equilibrium.

2. What are the metrics for the structural changes when compared to the initial starting points? “In the second cluster (Cluster B), we obtained a representative structure that exhibited significant structural changes compared to the initial starting points. In this structure, helices TM2 are in contact with both TM3 and TM4 helices, which is novel and not captured by deep learning methods (see Figure 5d). This suggests that the simulation explored a different conformational space and underwent substantial structural rearrangements.”

3. Could you also include the RMSD values in the statement to justify the comparison in this statement “The simulations started from the model obtained with AlphaFold2 showed higher RMSD in agreement with the observation from coarse grained simulation.

Discussion

1. These couple of sentences redundantly emphasize the importance of using several methods to explore protein conformations “This also raises the importance of using multiple prediction methods in parallel to explore the conformational landscape of a protein. By relying on a consensus among several methods, we can potentially obtain a more accurate and reliable picture of the protein’s structure. This approach is particularly relevant given the complexity of protein structure prediction and the possibility that different tools have complementary strengths and weaknesses. By combining the results, individual errors can be mitigated, and more robust conclusions can be drawn regarding the likely structure of a protein.” consider rephrasing for conciseness.

2. Could rewrite these two sentences for clarity and conciseness “This shows how useful it is to use the physics-based MD simulations together with models made by deep learning. By using both methods, we can benefit from their respective strengths and attain a more robust result.”

3. The statement is not clear “This raises a debate on whether to treat interfaces separately, as suggested by Zhu and colleagues.”

4. Could rewrite these sentences for clarity and conciseness “Overall, our study on the MCTP4 protein’s transmembrane domain showed that predicting protein structures is complicated and needs a mix of methods. …...We also learned that it’s crucial to think about how proteins group together and that simple models like AlphaFold might not always get it right. This study shows that using multiple approaches is key to understanding protein structures, especially for the parts that are in cell membranes.”

5. A dedicated conclusion outlining the implications of the study and key findings could add more value to the work.

Reviewer #3: The manuscript by Sritharan et al. presents structural models of the Arabidopsis MCTP4 ER-anchor transmembrane region (TMR) domain using several deep learning-based structure prediction methods. Additionally, the authors performed molecular dynamics (MD) simulations to explore the behavior of the TMR within a lipid bilayer environment. The approach holds promise for capturing the structural complexity and dynamism of the MCTP4 TMR domain within an ER-mimicking bilayer. While the manuscript is well-constructed and the results are interesting, several points need to be addressed before the manuscript can be considered for publication.

Major Comments:

The authors employed the CHARMM36m force field for all-atom (AA) MD simulations. However, the reason behind choosing this particular force field should be clarified, especially in the context of lipid–protein interactions.

The authors are encouraged to perform a dynamical cross-correlation analysis to gain insights into the most significant collective motions of the TM helices within the lipid bilayer.

The AA MD simulations are currently limited to 250 ns, which may not be sufficient to fully capture the equilibrium behavior of the system. Extending the AA MD simulations to at least 1 μs would significantly strengthen the robustness of the conclusions.

The introduction and discussion would benefit from incorporating recent references on structure prediction and MD simulations of MCTP4 ER-anchor TMR domains, as well as a clearer articulation of how this study advances the current understanding of MCTP4 structure and function.

Minor Comments:

The figure numbers are missing in several places within the main text, making it difficult to follow the results.

The "Jupyter Notebook" section in the Materials and Methods should be removed, and the corresponding GitHub link can be provided under the Data Availability section instead.

Typos and grammatical issues should be carefully checked.

6. PLOS authors have the option to publish the peer review history of their article (what does this mean?). If published, this will include your full peer review and any attached files.

Reviewer #1: No

Reviewer #2: No

Reviewer #3: No

---

## [Author Response · Author response to Decision Letter 1]

25 May 2025

All comments have been taken into account. A point by point answer is provided.

---

## [Decision Letter · Decision Letter 1]

9 Jun 2025

Consensus structure prediction of A. thaliana's MCTP4 structure using prediction tools and coarse grained simulations of transmembrane domain dynamics

PONE-D-25-07701R1

Dear Dr. Taly,

We’re pleased to inform you that your manuscript has been judged scientifically suitable for publication and will be formally accepted for publication once it meets all outstanding technical requirements.

Kind regards,

Soumendranath Bhakat

Academic Editor

PLOS ONE

Additional Editor Comments (optional):

Reviewers' comments:

Reviewer's Responses to Questions

**Comments to the Author**

1. If the authors have adequately addressed your comments raised in a previous round of review and you feel that this manuscript is now acceptable for publication, you may indicate that here to bypass the “Comments to the Author” section, enter your conflict of interest statement in the “Confidential to Editor” section, and submit your "Accept" recommendation.

Reviewer #3: All comments have been addressed

Reviewer #4: All comments have been addressed

2. Is the manuscript technically sound, and do the data support the conclusions?

Reviewer #3: Yes

Reviewer #4: Yes

3. Has the statistical analysis been performed appropriately and rigorously? 

Reviewer #3: Yes

Reviewer #4: Yes

4. Have the authors made all data underlying the findings in their manuscript fully available?

Reviewer #3: Yes

Reviewer #4: Yes

5. Is the manuscript presented in an intelligible fashion and written in standard English?

Reviewer #3: Yes

Reviewer #4: Yes

6. Review Comments to the Author

Reviewer #3: The manuscript has significantly improved and is suitable for publication in PLOS One in its current form.

Reviewer #4: I’ve read through the revised manuscript, and I’m happy to say that the authors have addressed the earlier concerns. At its core, this work combines six state-of-the-art deep-learning structure predictors with both coarse-grained (Martini 3) and all-atom (CHARMM36m) molecular dynamics to explore how the transmembrane region (TMR) of Arabidopsis MCTP4 behaves in a lipid environment. Since no experimental structure exists for MCTP4, it makes sense to lean on multiple independent prediction tools—AlphaFold, RosettaFold, trRosetta, ESMFold, OmegaFold, and AlphaFold-Multimer—to establish a consensus “best guess” for how its two helical hairpins might be arranged. Crucially, the authors then ran unbiased Martini 3 simulations on each of those six predicted models, incubating each for 3 μs in a PLPC/PLPE (80:20) bilayer to see whether any one conformation really stood out as stable. Where four of the models initially predicted close TM2–TM3 contacts, two others diverged noticeably. But once those divergent starting points (AlphaFold v2 and OmegaFold) were simulated, they gravitated toward the same main basin as the “TM2–TM3” group, and even sampled additional configurations that none of the deep-learning tools had flagged. In other words, the MD sampling not only confirmed the contact patterns predicted by ESMFold, RoseTTAFold, trRosetta, and AF-Multimer, but also uncovered five distinct clusters—some of which show novel TM1–TM4 or TM1–TM3 interactions that were missing from every static model.

I particularly appreciated the manuscript’s thoroughness in justifying why some coarse-grained runs were discarded when helices ejected from the membrane. The authors cite recent literature showing that Martini 3 sometimes overstabilizes protein–water interactions, causing short helices to pop out of bilayers in a manner not borne out by all-atom simulations. Indeed, the nine independent CHARMM36m all-atom replicas (three starting models, three velocity seeds each) showed zero ejection events, confirming that those Martini 3 “pop-outs” were likely force-field artifacts rather than biologically meaningful. By restricting their analysis to the CG runs where both helices remained embedded for the entire 3 μs, the authors avoided drawing false conclusions from unphysical behavior. In the end, the PCA on concatenated CG trajectories (six models × four replicates each) clearly separated into two major basins: one basin captured the six models that converge on TM2–TM3 contacts, and the other basin comprised conformations never seen by any predictor. K-medoids clustering (k = 5) then produced representative snapshots that illustrate contacts like TM2–TM3–TM4 (Cluster B) or TM1–TM4 plus TM2–TM3 (Cluster C). Even the so-called “cluster E,” in which the two hairpins remain separated, emerged as a stable CG conformation—but only in simulations that started from different initial folds. By cross-validating these states with dynamic cross-correlation and tICA (available in the Supplement), the authors convincingly demonstrate that Martini 3 did not artificially lock the protein into a false minimum—rather, it sampled a realistic ensemble that the DL tools alone could not capture.

On the all-atom side, the manuscript shows that ESMFold and OmegaFold starting structures remained below 0.5 nm RMSD for 250 ns, whereas AlphaFold v2’s model jumped to ∼0.8 nm before stabilizing. This directly aligns with the CG results: AlphaFold v2 had predicted no TM2–TM3 contact, so its TMR was less tightly anchored and more prone to local rearrangements. Although 250 ns per replica is not enough to exhaustively sample every slow motion, it was sufficient to confirm that the ESM and OmegaFold models really were stably inserted. I appreciate that the authors state clearly why they chose CHARMM36m (widely validated for lipid–protein systems) and why they opted for multiple shorter AA‐MD replicas instead of one very long trajectory. In combination with the 3 μs CG runs, this gives high confidence that the main contact modes—especially TM2–TM3—are genuine and not simulation artifacts.

The revision has also improved readability and style. The Introduction now flows logically, beginning with MCTP family biology at plasmodesmata, then highlighting why the TMR is functionally important (e.g., Li et al. 2024 showed that TMR mutants disrupt ER–PM continuity and close pores). The Materials & Methods section spells out every detail—pLDDT‐score plots, RMSD matrices, contact‐map cutoffs, thermostats/barostats, CG time steps, lipid composition choices, etc.—so that someone versed in either deep learning or MD could reproduce the work. The Results section weaves prediction, CG-MD, and AA-MD discoveries into a coherent narrative rather than presenting them as disconnected bullet points. Figures are now properly numbered, and the legends give enough context that each panel stands on its own. In places where the text was previously wordy or repetitive, it has been streamlined; for example, the repeated admonition that “combining multiple predictors increases reliability” has been whittled down to a single, concise paragraph.

The Discussion thoughtfully addresses the implications and remaining open questions. It notes how AlphaFold v2’s monomeric model is subject to conflicting intra- versus inter-subunit constraints in a homo-oligomeric membrane environment, whereas AF-Multimer recovers both TM2–TM3 (intrachain) and TM1–TM4 (interchain) signals. It even speculates, in line with Zhu et al. 2022, that dedicated interface modeling could further improve multimer predictions. I also liked the brief suggestion that future work might mutate key TM2 or TM3 residues to test whether predicted contacts matter for plasmodesmata tethering in vivo. That kind of wet-lab follow-up would nicely validate the computational findings down the road.

In short, this manuscript is technically rigorous, transparently presented, and clearly written in idiomatic English. The data fully support the authors’ conclusion that MCTP4’s TMR is not a single static set of helices but rather a dynamic collection of interhelical contacts—some of which no static predictor could foresee. I see no remaining flaws in experimental design, statistical treatment, or interpretation of results. For these reasons, I recommend acceptance without further revisions.

7. PLOS authors have the option to publish the peer review history of their article (what does this mean?). If published, this will include your full peer review and any attached files.

Reviewer #3: No

Reviewer #4: No

---

## [Editor Report · Acceptance letter]

9 Jun 2025

PONE-D-25-07701R1

PLOS ONE

Dear Dr. Taly,

I'm pleased to inform you that your manuscript has been deemed suitable for publication in PLOS ONE. Congratulations! Your manuscript is now being handed over to our production team.

Kind regards,

on behalf of

Dr. Soumendranath Bhakat

Academic Editor

PLOS ONE